# Modeling for understanding and engineering metabolism

Jens Nielsen[1,2,3,4] (ID) and Dina Petranovic[2,3]

[1]BioInnovation Institute, Copenhagen, Denmark; [2]Department of Life Sciences, Chalmers University of Technology, Gothenburg, Sweden; [3]Novo Nordisk Foundation Center for Biosustainability, Technical University of Denmark, Lyngby, Denmark and [4]Novo Nordisk Foundation Center for Basic Metabolic Research, University of Copenhagen, Copenhagen, Denmark

metabolic engineering; synthetic biology; systems biology

**Corresponding author:**
Jens Nielsen;
Email: jni@bii.dk

## Abstract

Metabolism is at the core of all functions of living cells as it provides Gibbs free energy and building blocks for synthesis of macromolecules, which are necessary for structures, growth, and proliferation. Metabolism is a complex network composed of thousands of reactions catalyzed by enzymes involving many co-factors and metabolites. Traditionally it has been difficult to study metabolism as a whole network and most traditional efforts were therefore focused on specific metabolic pathways, enzymes, and metabolites. By using engineering principles of mathematical modeling to analyze and study metabolism, as well as engineer it, that is, design and build, new metabolic features, it is possible to gain many new fundamental insights as well as applications in biotechnology. Here, we present the history and basic principles of engineering metabolism, as well as the newest developments in the field. We are using examples of applications in: (1) production of protein pharmaceuticals and chemicals; (2) basic studies of metabolism; and (3) impacting health care. We will end by discussing how engineering metabolism can benefit from advances in artificial intelligence (AI)-based models.

## Introduction

Metabolism is at the core of all cellular functions providing not only the energy equivalents for driving all chemical processes but also the building blocks for all cellular structures, growth, and proliferation. The metabolism is a network of many interconnected chemical reactions that in a coordinated fashion ensure the degradation and modification of nutrients, that is, from carbon and nitrogen sources, into different chemicals that the cells can use for the synthesis of building blocks like amino acids, fatty acids, and nucleotides. These building blocks are subsequently assembled into macromolecules like proteins, lipids, DNA, and RNA that make up the cell. Major breakthroughs in our understanding of metabolism were made at the beginning of the 20th century when Otto Meyerhof, Gustav Embden, and Jakub Karol Parnas identified the individual chemical reactions involved in the conversion of glucose to pyruvate. This pathway is today known as the Embden-Meyerhof-Parnas (EMP) pathway, but also often referred to as glycolysis. In 1922, Otto Meyerhof received the Nobel Prize in Physiology or Medicine for his work on mapping the conversion of glucose to lactic acid in muscle cells deprived of oxygen, so-called fermentative metabolism. In 1933, Otto Warburg received the Nobel Prize in Physiology or Medicine for his discovery of respiratory metabolism, but his name is today more known for the "Warburg effect" and his discovery that cancer cells tend to bypass respiration, in the presence of oxygen, as they convert glucose to lactate. This observation was also made by Herbert Crabtree in studies of cancer cells, but his name is today associated with the same phenomenon in yeast that produces ethanol at high glucose concentrations even in the presence of oxygen, the so-called "Crabtree effect" In 1953, Hans Krebs received the Nobel Prize for identifying the steps involved in the tricarboxylic acid (TCA) cycle, also often referred as Krebs cycle, where acetyl-CoA is degraded in a metabolic cycle to carbon dioxide resulting in the production of "energy equivalents". Fritz Lipmann shared the Nobel Prize in 1953 with Hans Krebs for identifying acetyl-CoA as a crucial molecule in linking the EMP pathway with the TCA cycle, via the enzyme Pyruvate Dehydrogenase. In 1978 Peter Mitchell received the Nobel Prize in chemistry for proposing the chemiosmotic theory, which describes how electrons are transferred from the co-factor NADH, generated in the TCA cycle, to oxygen reducing it to water. According to this theory, protons are being pumped against a concentration gradient out of the mitochondrial matrix when electrons transfer through the so-called electron transport chain. Paul D. Boyer and John E. Walker, who received the Nobel Prize in Chemistry in 1997 for the identification of the mechanisms of ATP Synthase, which forms ATP when protons re-enter the mitochondria, together with Jens C. Skou, who discovered the Na$^+$-K$^+$-ATP pump, and hereby all the mechanisms involved in both oxidative and fermentative metabolism of glucose had been mapped. Other Nobel Prizes in Chemistry were given for the discovery of specific metabolic pathways, that

is, to Lord Todd in 1957 for his discovery of the biosynthesis of nucleotides, to Melvin Calvin in 1960 for his discovery of the mechanisms behind carbon dioxide fixation, and to Luis Leloir in 1970 for his discovery of the role of sugar nucleotides in sugar metabolism, for example, for the metabolism of galactose, now known as the Leloir pathway. Many other Nobel Prizes have been given for discoveries of vitamins and other bioactive chemicals such as molecules, enzymes, and pathways of the so-called secondary metabolism, for example, terpenes, alkaloids, carotenoids, antibiotics, etc.

The more than 100 years of research on metabolism have resulted in extensive mapping of most core reactions engaged in energy generation and cell synthesis, but we don't always know the identity and characteristics of the enzymes carrying out the reactions. There are also many dark spots in certain parts of metabolism, especially in plants as their cells have evolved the capability to perform advanced chemistry, but modern molecular biology techniques have enabled the transfer and assembly of whole plant pathways into a host organism, for example, Baker's yeast, and this has enabled mapping of pathways leading to complex natural products. This type of heterologous expression assembles all the enzymes of many different biosynthetic pathways, even from different organisms, leading to the creation of various complex natural products in so-called cell factories (Gelanie *et al.*, 2015; Zhao *et al.*, 2023). Information about metabolic reactions and their associated enzymes is available in several different databases, for example, Kyoto Encyclopedia of Genes and Genomes (KEGG), which is a valuable resource that significantly advanced our ability to understand and thus engineer metabolism.

Several studies, including the seminal work related to developing Metabolic Control Analysis (Kacser and Burns, 1973; Kell *et al.*, 1989), have shown a big difference between enzymes operating in isolation (test tube, *in vitro*) and how enzymes operate within a pathway (in a cell, *in vivo*). Gaining insight into how sets of enzymes interact in a cell can therefore only be obtained through the introduction of mathematical models and in engineering disciplines, mathematical models play an important role in the design and development of new or improved products and processes and for analyzing data. Biological systems are inherently hard to engineer and describe mathematically using different models, especially as we do not know most of the components of the system, their roles and functions, as well as inherent redundancies, inefficiencies, and complexity. Even for the simplest cell we still have a challenge to know all the "parts", their individual functions and how they interact, as well as the unique features of that system. However, as we progress towards a complete overview of metabolism it is possible to build comprehensive models with remarkably wide applications for studying and designing cellular metabolism, with the objective of developing new cell factories, for biomarker discovery, drug discovery, and design of healthy diets for humans.

## History of engineering metabolism

In 1973 Herbert W. Boyer and Stanley N. Cohen invented genetic engineering, which laid the foundation for a multibillion-dollars industry, underlying everything we do today in modern molecular biology, both in academia and industry. Through the expression of a heterologous gene in a host cell like the bacterium *Escherichia coli* or Baker's yeast *Saccharomyces cerevisiae* their invention enabled scalable production of human proteins like insulin and growth hormone (Nielsen, 2013). Following the successful with expression of a single gene encoding a protein of commercial interest, genetic engineering was also exploited for expressing heterologous

enzymes to reconstruct heterologous biosynthetic pathways in various hosts. In 1991 this led to the coining of the term Metabolic Engineering by James E. Bailey and Gregory N. Stephanopoulos (Bailey, 1991; Stephanopoulos and Vallino, 1991), and over the last 30 years Metabolic Engineering has been established as an active research field with a dedicated textbook (Stephanopoulos *et al.*, 1998), dedicated journals, and several conference series.

Despite many successes in using Metabolic Engineering to develop new cell factories (see below), it has often been difficult to meet the techno-economic requirements for establishing commercial processes (Konzock and Nielsen, 2024). A major reason for this has been that it is often difficult to engineer the host metabolism in a way that ensures the conversion of most of the carbon atoms from the substrate towards the production of the chemical/compound of interest. The main reason for this is the trade-off between growth and product formation where most microorganisms have evolved to maximize growth. Re-directing flux towards the product of interest is therefore difficult for two main reasons: (1) metabolism is not organized into linear pathways but is a web (hairball) of interactions between metabolites and enzymes (Figure 1*a*), and the conversion of substrate to the product therefore often engages a very large number of reactions not directly involved in this conversion; and (2) flux through the different enzymes (Figure 1*b*) is extensively regulated at multiple levels, including the genome, transcriptome, proteome, and fluxome. This often results in the diversion of flux from the path between the substrate and the product. An approach of engineering specific enzymes at a time is therefore often failing, and even though automation has in recent years enabled rapid evaluation of many different engineering targets, it has become clear that a more holistic design approach needs to be applied, using mathematical models.

In 1979 Aiba and Matsuoka presented a simple model of citric acid production by the yeast *Candida lipolytica* (today renamed as *Yarrowia lipolytica*) and they were the first to use simple mass balancing around intracellular metabolites to calculate fluxes through metabolic pathways (Aiba and Matsuoka, 1979) (Figure 1*c*). In the 1980s and 90s, more mass-balance models of the central metabolism of various bacteria and fungi were made. Pioneers of developing bacterial models were Bernhard Palsson (Varma and Palsson, 1994) and Gregory N. Stephanopoulos (Vallino and Stephanopoulos, 1993), whereas Jens Nielsen pioneered the development of models for eukaryal organisms, that is, *S. cerevisiae* (Nissen *et al.*, 1997) and *Penicillium chrysogenum* (Jørgensen *et al.*, 1995). With the availability of genome sequences, it became possible to identify most of the enzymatic capabilities of a cell and hereby develop comprehensive mathematical models for metabolism. This led to the development of so-called genome-scale metabolic models (GEMs) for different bacteria by the Palsson group (Edwards and Palsson, 1999; Edwards and Palsson, 2000; Schilling *et al.*, 2002) and the first eukaryal cell (the yeast *S. cerevisiae*) by the Nielsen group (Förster *et al.*, 2003). *S. cerevisiae* is a widely used model organism in molecular and cell biology, as well as in industry where it is the most widely used cell factory for the production of food, beverages, chemicals, fuels, and pharmaceuticals (Nielsen, 2019). Its wide use as a model organism is well illustrated by the fact that several Nobel Prizes in Physiology or Medicine have been given to researchers who used yeast in their fundamental studies (Hohmann, 2016), for example, to Leland Hartwell, Paul Nurse, and Tim Hunt in 2001 for their discoveries of key regulators of the cell cycle, to James Rothman, Randy Scheckman, and Thomas Südhof in 2013 for their discovery of the protein

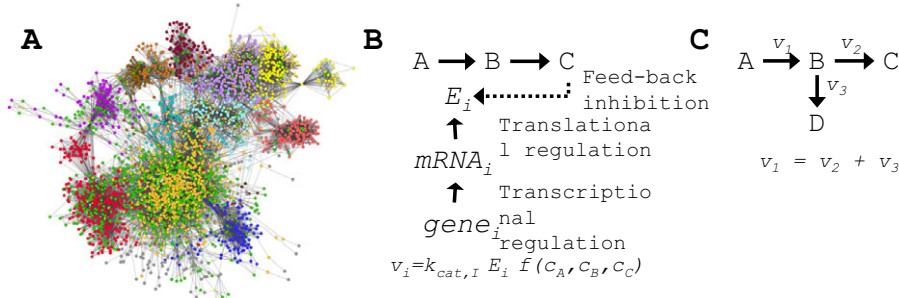

**Figure 1.** Metabolic networks and how their fluxes can be regulated. (a) Illustration of a typical hairball metabolic network. Green dots are enzymes and the dots with different colors are metabolites interacting with the enzymes. The metabolites are color-coded according to their cellular compartment. Most metabolites are in the cytosol (yellow dots) and the mitochondria (red dots). (b) Simple representation of the different layers of regulation of flux through a reaction that converts metabolite A to B. The reaction is catalyzed by the enzyme $E_i$ and the flux is a function of the catalytic capacity (turnover number) of this enzyme, that is, $k_{cat,I}$, the concentration of the enzyme ($E_i$), and a function of the different metabolites in the network (*f*). In the simple model, there is feedback inhibition of the enzyme by metabolite C, and the flux is therefore determined by the concentration of the three metabolites A, B, and C. The enzyme concentration is determined by transcriptional regulation of the corresponding gene and by translational regulation of the corresponding mRNA. (c) Simple illustration of the concept of flux balancing. In this simple network, the three fluxes are constrained by a simple mass balance around the metabolite B. For most intracellular metabolites the turnover is so high that an assumption of steady state, resulting in the simple algebraic constraint equation, is reasonable. If the cells are experiencing a significant environmental change there will be a short period of time where the steady state assumption does not apply, but as the characteristic time constant for most metabolite concentrations, that is, the concentration of the metabolite divided by the flux through the metabolite, is in the order of seconds (rarely minutes), a new steady state level of the metabolites will rapidly be obtained. Therefore, the simple balance equation is in practice always valid.

secretory pathway in eukaryal cells, and Yoshinori Ohsumi in 2016 for elucidating the mechanisms for autophagy.

Following these initial models, the Nielsen group reconstructed GEMs for many other important microorganisms, such as *Streptomyces coelicolor* used for antibiotics production (Borodina *et al.*, 2005), *Aspergillus niger* used for the production of citric acid and many industrial enzymes (Andersen *et al.*, 2008), *A. oryzae* used for the production of many fermented food products and industrial enzymes (Vongsangnak *et al.*, 2008), and *P. chrysogenum* is used for the production of penicillin and other antibiotics (Agren *et al.*, 2013). Since this early reconstruction of metabolic networks there have been developed GEMs for many organisms (Gu *et al.*, 2019). Many models have been updated regularly as new biochemical information becomes available. For example, our yeast GEMs have had many different updates of the original model, with the most comprehensive model Yeast8 comprising almost 4,000 metabolic reactions linked to more than 1,100 enzymes and genes (Lu *et al.*, 2019). Yeast8 also represented a breakthrough in terms of engaging the scientific community as the model was made available open-source via GitHub. Hereby researchers from around the world could contribute with annotation, curation, and propose model updates and this has resulted in a recent update of the model (Yeast9), that contains information on 30 additional genes, 203 new reactions, and 140 new metabolites (Zhang *et al.*, 2024). These comprehensive models not only represent an extensive database of cellular metabolism, but also find application in metabolic engineering, basic science, and biomedical research as we will discuss below. Despite their scale, these models, still cover only the core of metabolism, as demonstrated in our discussion on using artificial intelligence to gainnew insights into metabolism.

## Engineering metabolism for industrial production

### Building a new (biotech) industry

The invention of genetic engineering resulted in the establishment of an industry for the production of recombinant proteins used as therapeutics exceeding USD100B (Nielsen, 2013). Human insulin was launched as the first product by Ely Lilly in 1982, with the second being human growth hormone launched by Genentech

in 1985. Today, there are more than 300 different recombinant proteins approved as therapeutics with many more in clinical trials, and among the top 10 selling pharmaceuticals 9 are biopharmaceuticals, which is a significant change from 20 years ago when most top-selling pharmaceuticals were small molecules.

Many different cell factories are being used for the production of recombinant proteins. The bacterium *E. coli* is used for small proteins such as interleukins and insulin, but even though this organism enables high expression levels a disadvantage is that the recombinant protein accumulates intracellularly in so-called inclusion bodies, which makes the purification process more cumbersome and costly. *S. cerevisiae* is characterized by having a fully functional protein secretion pathway, and it can therefore secrete recombinant proteins to the extracellular medium, which facilitates purification. This yeast is therefore widely used for recombinant protein production, for example, for insulin, human serum albumin, and glucagon-like peptide 1 receptor agonists (GLP1) (Huang *et al.*, 2014). The latter group of molecules mimics the action of the endogenous incretin hormone GLP1 that is released after eating, stimulating satiety. These molecules have therefore formed the basis for the active ingredient in the recently launched type-2 diabetes and obesity drugs Ozempic and Wegovy by Novo Nordisk.

The methylotrophic yeast *Komagataella phaffii* (formerly known as *Pichia pastoris*), is also a relevant cell factory as it ensures very high productivity due to a very efficient methanol-induced expression system and the ability for the yeast to grow to very high cell densities. We developed a GEM for *K. phaffii* (Caspeta *et al.*, 2012) and expanded this model to describe how metabolism could be engineered to enable humanized glycosylation of recombinant proteins (Irani *et al.*, 2015). This enabled the identification of metabolic engineering targets for improved production of a range of different recombinant proteins having industrial relevance. For the production of more complex proteins such as antibodies, erythropoietin, and blood coagulation factors, for example, Factor VII and Factor VIII, that require proper glycosylation it is necessary to apply a more advanced cell factory that can ensure that the protein is produced with the proper human glycosylation pattern. The Chinese Hamster Ovary (CHO) cells are often used but the original productivity of the CHO cells was relatively low and there were issues with strain stability. However, there has been significant

advancement in developing detailed mathematical models for CHO cells and our focus was to model the secretory pathway as this is crucial for ensuring efficient protein secretion by this cell factory (Gutierrez et al., 2020). At the same time, through advancement in ability to engineer CHO cells, it has been possible to improve productivity significantly, and CHO cells have therefore become the cell factory of choice for proteins that require proper human glycosylation.

As the secretory pathway is of such critical importance for many processes in engineering, we also focused on developing detailed mathematical models for the protein secretory pathway in *S. cerevisiae*. In one study the protein secretory pathway was completely mapped, with 162 proteins in this pathway engaged in processing 1,190 proteins (Feizi et al., 2013). Very few of the endogenous proteins being processed by the pathway are eventually secreted, whereas the remainder is processed through this pathway to be directed to the cell membrane where they function as transporters or receptors. Production of a heterologous protein will therefore compete with the capacity of this pathway, and if a heterologous protein is expressed at a very high level, it will therefore drain the capacity for processing membrane proteins, which will impact overall cellular functions. To quantify the demand for resources in the complex protein secretory pathway a detailed enzyme-constrained mathematical model was established (Li et al., 2022a), and using this model it was possible to quantify the impact of producing different recombinant proteins. It was further possible to use the model to design how the protein secretory pathway could be improved for the production of a specific protein, and many of these designs could be experimentally validated (Li et al., 2022a). Furthermore, the design strategies proposed by the model matched many earlier targets that had been identified through sequencing of different mutant strains with naturally varying secretion capacities (Huang et al., 2015; Huang et al., 2017; Huang et al., 2018).

### Transforming the old (chemical) industry

With the current wide use of microbial fermentation for the production of recombinant proteins used as advanced medicines, it is interesting to note that the use of microbial fermentation for the production of chemicals dates to the mid-19th century, when ethanol produced through yeast fermentation was used as a lighting fuel. In 1908 the demand for ethanol increased as it was used to fuel Henry Ford's Model T. With the oil boom gasoline took the lead, but in the 1920s and 1930s ethanol was used as an octane booster, and it was in high demand during World War II (WWII) due to fuel shortages. In the 1970s, with the rise in oil prices, ethanol was gaining renewed interest as a blend-in fuel, and this resulted in the establishment of the current very large ethanol industry; for example, in 2023 more than 110 billion liters of ethanol were produced through yeast fermentation. In 1908 there was another landmark in microbial production of chemicals, namely the development of acetone-butanol production by Chaim Weismann, a lecturer at Manchester University and later the first president of Israel, who developed a process based on fermentation with the bacterium *Clostridium acetobutylicum*. During World War I there was a large demand for acetone to be used in gunpowder. Acetone had earlier been produced from calcium acetate imported from Germany, and the Weismann process therefore became an important new route to obtain acetone for the United Kingdom. Production of ethanol and the Weismann process are both anaerobic processes, that is, there is no need for the provision of oxygen to

the cells. A key landmark was therefore the production of citric acid through fermentation with the filamentous fungus *Aspergillus niger*, which was introduced in 1919. This fungus is extremely tolerant to low pH, and the fermentation process can therefore be operated at pH 2–3, which reduces the demand for maintenance of aseptic conditions as very few other microorganisms can survive at these conditions. Following the discovery of penicillin by Alexander Flemming in 1928, there were extensive attempts to chemically synthesize this new bioactive to be used as an antibiotic, but during WWII it was decided to start fermentation-based production using the filamentous fungus *Penicillium chrysogenum*. This resulted in the development of the first aerobic fermentation process that required a supply of large amounts of aseptic air to the fermentation process, and the establishment of this process therefore laid the basis for the production of many different chemicals using microbial aerobic fermentation. Retrospectively it turned out to be a wise decision to choose the fermentation route for the production of penicillin as it was first demonstrated possible to chemically synthesize penicillin in 1956, and the chemical synthesis route cannot compete with the fermentation route. Today most antibiotics in the world are being produced by microbial fermentation.

Today microbial fermentation represents an industry exceeding USD100 billion, and many different chemicals are being produced through this route. Many new processes have been developed, and the application of Metabolic Engineering can be categorized into four main applications: (1) Engineering central metabolism to improve titers, rates, and yields (TRYs); (2) Extension of substrate range of the cell factory; (3) Improving tolerance of the cell factory; and (4) Expression of heterologous pathways for the production of valuable chemicals.

**Improvement of titers, rates, and yields** is essential for the establishment of an industrially viable process (Konzock and Nielsen, 2024). Even though it may be possible to extend endogenous pathways with a few enzymatic steps leading to the product of interest, the yield of product from the feedstock, typically glucose, and the rate of production, may often be low and it is therefore necessary to engineer the central carbon metabolism of the cell factory to ensure that carbon is directed from glucose to the product at a high rate. This is well illustrated in work on engineering *Escherichia coli* for the production of 1,3 propanediol (Nakamura and Whited 2003) and 1,4 butanediol (Yim et al., 2011). In both cases, the bacterium was heavily engineered with more than 10 different genetic modifications that ensured a high rate and yield. Today these two chemicals are being produced commercially and used in the production of various plastics. Lactic acid has been commercially produced through fermentation with lactic acid bacteria since the early 20th century for use in the food industry, but with the development of technology for polymerization of lactic acid to polylactate (PLA), that is a polymer with valuable properties such as biodegradability, good layer adhesion, and high strength, there was a need for large volumes of pure lactic acid and not the lactate salt. Lactic acid bacteria require supplementation of complex feedstocks, and this makes it expensive and difficult to obtain pure lactic acid from these fermentations. The company Cargill therefore developed a new process based on an engineered yeast that can tolerate low pH for the production of lactic acid, and this process resulted in a significant expansion of lactic acid production. More recently the company Corbion developed a process for purification of lactic acid from a traditional lactic acid bacterial fermentation process, and hereby enabled supply to production of PLA. Engineering of the central metabolism has also been shown to enable the improvement of ethanol production and reduction of production of

the by-product glycerol. Production of ethanol from glucose is completely balanced in terms of redox potential as there is also one molecule of carbon dioxide per molecule of ethanol formed, but part of the glucose is used for the production of more yeast cells that are more reduced than glucose, there is a need for an electron sink, and converting part of the glucose to glycerol represents such an electron sink (glycerol is more reduced than glucose). Cells use different co-factors to balance electron flows within metabolism, and based on metabolic modeling it was identified that by engineering pathways for ammonia assimilation, it should be possible to reduce the requirement for glycerol production and hereby increase ethanol production by 5–8%, which was experimentally validated (Nissen *et al.*, 2000). Also, mathematical modeling of the central carbon metabolism guided how changing the co-factor usage in glycolysis could give a similar effect (Bro *et al.*, 2006). GEMs have also played an important role in the identification of targets for engineering yeast for over-production and secretion of free fatty acids by engineering both the central carbon metabolism and the fatty acid metabolism (Zhou *et al.*, 2016). Through pursuing additional model-guided targets it was further possible to engineer the central carbon metabolism such that yeast could be transformed from alcoholic fermentation leading to ethanol production to efficient conversion of glucose to free fatty acids (Yu et al., 2018). As the conversion of glucose to free fatty acids is redox im-balanced the yield was, however, found to be relatively low, but through reconstruction of completely synthetic glycolysis identified through modeling it was possible to overcome this challenge and hereby an even better-producing strain was obtained (Qin *et al.*, 2023).

Many cell factories have a **limited range of carbon sources** they can use efficiently. Thus, yeast, the most widely used cell factory, is not very efficient in using galactose and it cannot use xylose and arabinose as carbon and energy sources. This is important in the context of using lignocellulosic materials as feedstock for the production of fuels and chemicals, as galactose, xylose, and arabinose are abundant sugars in these feedstocks. Galactose is metabolized via the Leloir pathway, and even though this pathway only involves a few additional steps compared with the metabolism of glucose, the pathway is quite inefficient, and the growth rate of yeast on galactose is less than 50% compared with glucose (Ostergaard *et al.*, 2000). The Leloir pathway is tightly regulated, but through engineering the regulatory machinery rather than simply over-expressing the enzymes it was possible to significantly increase the growth rate of yeast on galactose (Ostergaard *et al.*, 2000). Using mathematical modeling we later found that this was due to a requirement for balanced expression of the individual enzymes in the pathway as the pathway intermediates can inhibit enzymes in the pathway (de Jongh *et al.*, 2008). This led to the identification of a downstream enzyme, phosphoglucomutase (Pgm2) that converts glucose-1-phosphate to glucose-6-phosphate and has traditionally not been considered part of the Leloir pathway, as a flux controlling enzyme, and by over-expressing this single enzyme it was also found possible to significantly increase the specific growth rate (Bro *et al.*, 2005). In a later study where we used adaptive laboratory evolution, we found that by combined over-expression of Pgm2 and engineering of a pathway engaged in glucose regulation, it was possible to further enhance the growth rate on galactose as this ensured balancing of flux through all the individual reactions and therefore no accumulation of intermediates (Hong *et al.*, 2011). Engineering yeast to efficiently use xylose and arabinose has been attempted since the 1980s, but the discovery of a fungal xylose isomerase that converts xylose to xylulose, that yeast can metabolize, in a single step represented a breakthrough (Kuyper *et al.*,

2005). This also opened for engineering yeast to use arabinose (Wisselink *et al.*, 2007). However, yeast has not evolved to use xylose efficiently and the central carbon metabolism is therefore not well balanced to handle this carbon source. Using mathematical modeling to guide extensive engineering of the central carbon metabolism, it was possible to pinpoint key targets among more than 100 different genetic modifications, and hereby engineer yeast metabolism to efficiently grow on xylose as the sole carbon source (Li *et al.*, 2021).

Tolerance towards environmental stress is an important feature of cell factories, and it is therefore interesting to find strategies to **improve tolerance** of cell factories towards chemicals present in the medium, chemicals produced by the cell factory, low/high pH, and low/high temperature. Often a cell factory is chosen for its tolerance towards specific environmental conditions, for example, *A. niger* is a well-suited cell factory for the production of citric acid as it can tolerate low pH and *S. cerevisiae* is a well-suited cell factory for the production of ethanol as it is very tolerant towards this chemical. However, in the current ethanol industry, it is desirable to operate at a higher temperature than the optimum temperature for *S. cerevisiae*, but the biology associated with ensuring temperature tolerance is not known in detail. Here the concept of adaptive laboratory evolution was shown to be efficient as it enabled, through a sequential selection of yeast clones, to identification of clones that could grow faster at elevated temperatures, that is, 40°C versus the optimum for the yeast of 35°C (Caspeta *et al.*, 2014). Through genome sequencing combined with metabolic modeling of isolated clones, it was then possible to identify causal mutations, and in particular, one mutation was identified to cause a loss of function of an enzyme involved in the biosynthesis of the membrane component ergosterol. The loss of function of this enzyme caused the production of another sterol, namely fecosterol, and this resulted in a slightly stiffer membrane property that could enable the cells to better function at higher temperatures (Caspeta *et al.*, 2014). A similar approach has been applied to make yeast more tolerant to lactic acid at low pH (Fletcher *et al.*, 2016) and more tolerant to dicarboxylic acids (Pereira *et al.*, 2019), but it has also been used to improve the tolerance of *E. coli* to various toxic chemicals (Lennen *et al.*, 2023).

Finally, engineering metabolism has been used to recruit different cell factories such as yeast for the expression of **heterologous pathways**, hereby enabling the production of complex natural products. Thus, yeast has been recruited to produce complex plant chemicals such as opioids (Gelanie *et al.*, 2015), monoindole alkaloids that can be used as anti-cancer drugs (Zhang *et al.*, 2022), celastrol that has anti-obesity properties (Zhao *et al.*, 2023), berberine that has anti-diabetic properties (Jiao *et al.*, 2024), polyamines and polyamine conjugates (Qin *et al.*, 2021), and isoflavonoids such as puerarin and daidzin that have cardioprotective properties (Liu *et al.*, 2021). Yeast is a well-suited cell factory for the production of these complex plant natural products as biosynthesis of many of these molecules involves complex oxidation reactions catalyzed by so-called P450 enzymes. These enzymes contain a heme group in the catalytic core and both heme and iron can often become limited for the function of these enzymes. However, through the engineering of the heme biosynthetic pathway, it was shown possible to enable elevated activity of P450 enzymes (Michener *et al.*, 2012). By expanding an enzyme-constrained yeast GEM to include the incorporation of metals it was possible to predict the effect of iron supplementation on the biosynthesis of coumaric acid, which involves a single P450 enzyme (Chen *et al.*, 2021). Hereby it was shown how GEMs can enable the advancement of all the efforts on

engineering yeast for the production of natural products towards commercial production.

The above examples show that even though there have been many successful examples of engineering metabolism without the guidance of mathematical models, these models have today come to a stage where they can significantly impact cell factory design (Domenzain *et al.*, 2024). The field is therefore aligning more with traditional engineering disciplines where mathematical models are actively used in design, and the design-build-test-learn cycle is therefore becoming well integrated into biology research (Nielsen and Keasling, 2016). Here it is interesting to note that even though cell factory development is generally carried out with a clear objective, that is, to develop a cell factory that has improved properties, analysis of newly engineered cell factories often results in a new understanding of cellular metabolism, and the border between engineering and basic biological research is therefore disappearing.

## Systems biology of metabolism

Systems biology is the mathematical analysis and modeling of complex biological systems. It has two historical roots (Westerhoff and Palsson, 2004): (1) from theoretical biology, and (2) from genome-sequencing. The root from theoretical biology is often referred to as a bottom-up approach as it is based on detailed mathematical modeling of specific molecular processes whereas the root from genome-sequencing is often referred to as a top-down approach (Nielsen, 2017). Theoretical biology developed together with a new understanding of molecular mechanisms in biology, as illustrated by the classical discovery of the mechanism on how the expression of genes encodes enzymes is involved in the metabolism of lactose in *E. coli*, present in the so-called Lac-operon. This genetic system was discovered by Francois Jacob and Jacques Monod, who received the Nobel Prize in Physiology or Medicine in 1965. The molecular understanding of the Lac-operon has formed the basis for the development of detailed mathematical models (Lee and Bailey, 1984), and these pioneering studies led to an expansion of the field of mathematical models of biological systems. However, most of these models only describe a specific process within the cell and do not capture overall cellular metabolism. There have been attempts to develop whole-cell models, for example, a comprehensive model for *E. coli* (Karr *et al.*, 2012), but these models are often relying on descriptions of several processes through empirical expressions rather than detailed molecular, mechanistic models. GEMs are in principle a merger of bottom-up and top-down

approaches as these models are capturing individual enzymatic reactions, but by doing it genome-scale the model is relying on genome information. The models do, however, deviate from many mechanistic bottom-up models as they do not rely on mechanistic models describing the kinetics of each enzymatic reaction, and the models therefore only have very few parameters that need to be estimated based on fitting to experimental data. This feature makes them attractive for wide use as it is hereby relatively easy to build models based only on the genome sequence and stoichiometry of reactions.

Whereas the first GEMs were reconstructed in a bottom-up fashion, the availability of GEMs from many different organisms and extremely well-curated models like Yeast8 (Lu *et al.*, 2019), it has become possible to almost automatically generate GEMs for different organisms, that is, more than 350 different yeast species were reconstructed by using Yeast8 as a template (Lu *et al.*, 2021). These models were used to gain insight into many different aspects of fundamental knowledge of metabolism and evolution, that is, how metabolism has evolved in different yeast species to adapt to various ecological niches. A similar approach was taken in using the well-curated GEM for human metabolism, Human1, as a template model for generating GEMs for different model organisms like mice, rats, zebrafish, fruit flies, and nematode (Wang *et al.*, 2021). The hereby generated GEMs were used for the identification of differences and commonalities in terms of metabolism between these very diverse species that are all used as model organisms for studying the biology of human cells, tissues, and organs. Specifically, our mouse GEM was used to analyze data from mice to gain new insight into how the development of Alzheimer's disease is associated with dramatic metabolic alterations in neuronal cells (Wang *et al.*, 2021).

Even though GEMs have a remarkable predictive strength, it is required to impose key constraints on the model for predicting a relevant phenotype. For example, to predict the growth rate of a cell it is necessary to constrain the nutrient uptake rate or vice versa (Figure 2a). Furthermore, GEMs can predict very large fluxes through pathways that in practice may have very little capacity, either due to low concentrations of the enzymes or due to low catalytic efficiency of the enzymes, that is, low enzyme turnover numbers. To overcome this problem we developed the concept of enzyme-constrained GEMs (ecGEMs) using the modeling framework we call GECKO (Sanchez *et al.*, 2017). In ecGEMs, the flux through each individual enzyme is constrained by the turnover number and the enzyme concentration (Figure 2b). This enables

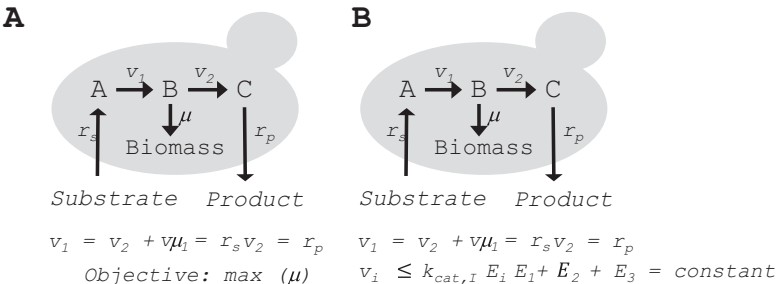

**Figure 2.** Constraints of GEMs and how they impact flux estimation. (a) In simple flux balance analysis where model simulation is based on the mass balances illustrated in Figure 1c it is necessary to constrain either one input or one output flux combined with an objective function, here illustrated by maximizing the specific growth rate μ. This is often one of two options: (1) the substrate uptake rate is defined and there is optimized for growth rate, or (2) the growth rate is defined and there is minimized for substrate uptake rate. If $r_s$ is given and there is maximized for μ then the model will obviously not predict any product formation, that is, $r_p$ is zero. (b) By constraining the flux through each of the reactions by the enzyme turnover number ($k_{cat}$) and the enzyme concentration ($E_i$) it is not necessary to constrain any input or output flux and the model therefore has better predictive strength. Either the enzyme concentrations can be given as input, or the sum of all enzyme concentrations is capped at a constant value that is consistent with experimental measurements.

significant improvement in the predictive strength of these models, both in terms of flux distribution, growth on different carbon sources, and prediction of overflow metabolism, that is, the Crabtree effect (Sanchez *et al.*, 2017). A trade-off with these models is, however, that they require information about the turnover numbers for all the enzymes, that is, $k_{cat}$'s, and information about the enzyme levels. For well-studied organisms like *S. cerevsisiae* and *E. coli*, there are extensive databases, for example, BRENDA, of $k_{cat}$ values, and if there are no values available for some enzymes $k_{cat}$ values determined for similar enzymes in other organisms can be used as default. For many well-studied enzymes, there are even several different $k_{cat}$ values reported, and in some cases, these values even vary by an order of magnitude. In these cases, GECKO selects the largest values in order not to over-constrain flux through the reaction (Sanchez *et al.*, 2017). The GECKO framework has been further developed to automatically sample $k_{cat}$ values from databases, and this has enabled faster reconstruction of ecGEMs for different microorganisms (Domenzain *et al.*, 2022). There is less information about the enzyme concentrations, and even though quantitative proteomics has advanced significantly, in particular for *S. cerevisiae* (Lahtvee *et al.*, 2017; Yu *et al.*, 2020; Di Bartolomeo *et al.*, 2020), it is still laborious to obtain high-quality proteome data, but in these cases there can be used a constraint about a total proteome allocation to metabolic enzymes (Figure 2*b*), and it turns out that this allocation is remarkable constant across strains and different environmental conditions (Sanchez *et al.*, 2017).

ecGEMs have been shown to have strong predictive strength, most likely as they are rooted in a biological constraint that is deeply rooted in evolution, namely a constraint on protein synthesis rate by the ribosomes. For many organisms, there is a linear correlation between ribosomal RNA content and specific growth rate, and quantitative proteomics has confirmed this for ribosomal protein content (Xia *et al.*, 2022). If the cell can reduce the proteome required for a certain part of metabolism, for example, for the biosynthesis of amino acids if these are supplied to the medium, this proteome mass can be allocated to ribosomes, and hereby the cell can grow faster (Björkeroth *et al.*, 2020). This clearly shows that proteome allocation within the cell is important and therefore also imposes overall constraints on metabolism. ecGEMs can therefore describe a phenomenon that has been a conundrum for many years, why do fast-growing cells use metabolic pathways that provide less energy per unit glucose, that is, overflow to ethanol in yeast cells (the Crabtree effect), to acetate in *E. coli* cells and to lactic acid in human cells, instead of using complete respiration that results in the extraction of far more energy. In fact, ecGEMs can very well describe this overflow metabolism (Sanchez *et al.*, 2017), and using quantitative proteomics data it has been shown that the underlying reason is that overflow metabolism is more efficient in terms of energy not per glucose but **per proteome mass of the energy generating pathway** (Chen and Nielsen 2019). Thus, in nutrient excess, at fast growth, the cells have evolved to prioritize proteome allocation towards ribosomes rather than efficient energy generation, where fully respiratory metabolism is "proteome-expensive".

GEMs are an excellent platform for integrative analysis of so-called -*omics* data, that is, transcriptome, proteome, and metabolome data. As these models are comprehensive in terms of covering all enzymes in the metabolic network of a cell and as there is a direct link to the encoding genes it is possible to directly overlay different -*omics* data onto the metabolic networks. More importantly, it is even possible to combine this with statistical methods for the identification of what we define as so-called reporter metabolites (Figure 3*a*) (Patil and Nielsen, 2005; Oliveira *et al.*, 2008). These

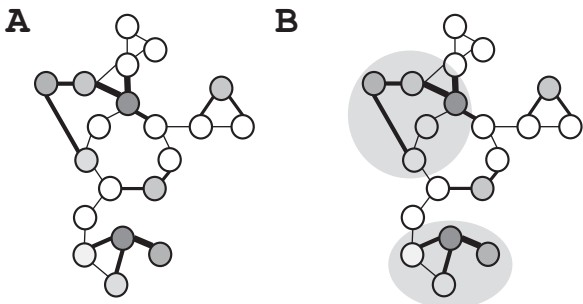

**Figure 3.** Use of GEMs for integrative analysis of omics data. (a) Using the graphical structure of GEMs it is possible to identify Reporter Metabolites, which are metabolites in the metabolic network around which there are significant changes in transcript level or protein level. The lines represent enzyme-catalyzed reactions and the circles are metabolites. The thickness of the lines indicates the changes in enzyme levels, measured by transcripts or protein levels. Metabolites around which there are large changes in enzyme levels become reporter metabolites and the significance is marked by the greyness, with dark grey being very significant and light grey less significant. (b) The graphical structure of GEMs can enable the identification of Reporter Networks, which are sub-networks where there are significant changes in the transcription level or protein level. Two sub-networks are marked with light grey circles.

are metabolites in the metabolic network for which there are significant alterations in the expression of the enzymes that engage in chemical reactions with the metabolite, either using the metabolite as a substrate or producing the metabolite. Altered expression can be quantified by measuring gene expression, for example, through mRNA sequencing, or through quantitative proteomics, and reporter metabolites therefore identify spots within the metabolic network where there are altered enzyme levels either to maintain homeostasis of metabolism or to drive a metabolic change. With the presence of high-quality metabolomics data, it is also possible to identify reporter enzymes, that point to key enzymes involved in handling altered metabolite levels within the metabolic network (Cakir *et al.*, 2006). The concept of reporter metabolites has gained much traction and led to the development of PIANO, our platform that enables easy analysis of different types of I data (Väremo *et al.*, 2013).

## Understanding metabolism for human health

GEMs have also been developed for human cells (Duarte *et al.*, 2007; Ma *et al.*, 2007; Mardinoglu *et al.*, 2013) and the consensus model Human1 was established using the same concept as we used for building the consensus Yeast8 (Robinson *et al.*, 2020). These models have been used extensively for mapping metabolic changes associated with disease development, for example, for analyzing the metabolic changes in adipocytes in response to obesity (Mardinoglu *et al.*, 2013) and how liver metabolism becomes serine deficient in response to development of non-alcoholic fatty liver disease (NAFLD) (Mardinoglu *et al.*, 2014). GEMs have also been used to analyze metabolism during high-intensity exercise (Nilsson *et al.*, 2019). Besides enabling a new understanding of how metabolism responds to disease development, human GEMs find wide applications for biomarker identification and drug discovery.

Identification of reporter metabolites has shown to be very useful for **biomarker identification**. In a study of how metabolism is changing in 10 different cancers, we found that metabolism in the different cancers is more similar to the tissue of origin than among the different cancers (Gatto *et al.*, 2014). The metabolism in one cancer type, namely clear cell renal cellular carcinoma (ccRCC),

was found to be quite distinct, and through further analysis, a larger number of reporter metabolites were identified to be associated with glycosaminoglycan metabolism (Gatto *et al.*, 2014). Through measurements of 19 glycosaminoglycans in blood and urine in patients with metastatic ccRCC and healthy controls, a systems biomarker derived using machine learning from concentrations of the 19 metabolites could be identified (Gatto *et al.*, 2016). Further analysis showed that the systems biomarker could be used to detect early-stage solid tumors of ccRCC (Gatto *et al.*, 2018). These findings are now taken forward in clinical trials to validate a biomarker approved for the detection of recurrence in ccRCC patients from both urine and blood (clinical study AURORAX-0087A; NCT04006405). We also found that the systems biomarker has been shown to have much wider applicability as it has been found possible to enable early detection of more than 14 different cancer types from a blood sample (Bratulic *et al.*, 2022). Cancer detection using liquid biomarkers has become attractive with a potential future market exceeding USD10B, much thanks to the demonstration of early cancer detection through deep sequencing of DNA in blood samples (Jamshidi *et al.*, 2022). However, the systems biomarker based on measurements of glycosaminoglycans is attractive due to its lower cost and its combination of high selectivity and sensitivity (Bratulic *et al.*, 2022).

GEMs can also be used for drug discovery as was illustrated for both cancer and disrupted metabolism associated with aging (Folger *et al.*, 2011; Yizhak *et al.*, 2013). In one study we found that fatty acid oxidation in mitochondria plays a central role in hepatocellular carcinoma (HCC), the most dominant form of liver cancer, and by blocking the transport of fatty acids to the mitochondria cancer cell growth could be blocked (Agren *et al.*, 2014). Another example is the identification of serine deficiency in patients with NAFLD leading to the use of our GEMs for drug discovery: serine deficiency, together with other metabolic alterations, led to proposing a cocktail of metabolites, that is, serine, L-carnitine, nicotinamide riboside and N-acetyl-L-cysteine, that was found in clinical trials to aid the treatment of NAFLD (Zeybel *et al.*, 2021) and of mild-to-moderate COVID-19 (Altay *et al.*, 2021). The cocktail has also passed phase 2 in clinical trials for improving cognitive function in patients with Alzheimer's Disease (Yulug *et al.*, 2023). In the cocktail serine and N-acetyl-L-cysteine serve as precursors for biosynthesis of the important anti-oxidant glutathione, L-carnitine ensures efficient transport of fatty acids to the mitochondria for $\beta$-oxidation and nicotinamide riboside serves as a precursor for the co-factor nicotinamide adenine dinucleotide ($NAD^+$).

Besides our own human cells, tissues, and organs, we also have a very important metabolic organ composed of the human gut microbiome, which is a complex biological system comprising more than 1,000 different microorganisms, that communicate and exchange metabolites among each other and our human cells. We have about as many microbial cells as human cells in our bodies, so it is no surprise that the quality and composition of this microbiome have been shown to impact human health in different ways (Karlsson *et al.*, 2013; Ji and Nielsen, 2015; Schmidt *et al.*, 2018). Most studies in this field are associative, but an increasing number of studies have now identified causality between the human gut microbiome composition and disease development, for example, how the gut microbiome composition alters intestinal inflammation in colitis (Zhu *et al.*, 2018) and in the field of cancer treatment with immune therapies where we have shown that the presence of specific bacteria increases the response to treatment with checkpoint inhibitors (Limeta *et al.*, 2020). The gut microbiome composition evolves based on dietary intake, but due to extensive metabolic interactions between the many different microorganisms, it is difficult to predict how the composition exactly changes in response to diet. GEMs represent an excellent platform for the analysis of metabolic interactions between the many different bacteria and the host (Karlsson *et al.*, 2011), and through reconstructing GEMs for three dominant gut bacteria, we showed that it is possible to simulate bacterial interactions, and how the growth of the three bacterial species depends on the nutrients provided (Shoaie *et al.*, 2013). This concept further enabled simulation of how bacteria in the human gut microbiome contribute to amino acid biosynthesis in the human body, and hereby how the levels of amino acids change in the blood when human subjects undergo dietary interventions (Shoaie *et al.*, 2015). In this study, a group of overweight individuals was provided with a low-calorie diet for 6-weeks, and it was found that the response to the dietary change was dependent on the gut microbiome composition. Modeling showed that subjects with a less diverse microbiome responded better to dietary intervention, that is, their health status measured by several blood markers, including amino acid levels, than subjects with a very diverse microbiome (Shoaie *et al.*, 2015). The findings were later confirmed in a controlled mice study (Mardinoglu *et al.*, 2015). To predict how diet influences human gut microbiome composition, microbiome modeling was combined with a detailed mathematical model of the entire human gastrointestinal system, and this enabled for the first time simulation of how the gut microbiome evolves in infants when they change their diet from breast milk to solid food (Geng *et al.*, 2021). These modeling efforts of the human gut microbiome will enable better design of dietary interventions aimed at modulating microbiome. They will also support for the identification of new probiotics that help maintain a healthy gut microbiome composition thereby contribute to improved human health. Currently, the world market for probiotics exceeds USD50B and this is likely going to increase significantly in the future when it will be possible to design better products that have clinically validated health claims.

## AI for metabolism

Use of computational tools for the analysis and engineering of biological complex systems, such as metabolism is at a breaking point as standard mathematical modeling and GEMs have not fully solved the three major challenges in current biotechnology: (1) recombinant proteins used as pharmaceuticals (with an increasing market share of the total pharmaceutical market, which in itself is growing), are relatively expensive and it will be necessary to reduce production costs in order to make the drugs more widely available at fair price; (2) engineered microbes for production of chemicals and new products need not only new cell factories but also improved design processes in order to reduce the development costs (Nielsen *et al.*, 2022); and (3) finding biomarkers and drug targets for improving human health needs faster validation, and as new experimental platforms like human organoids are increasingly more useful there is a need for holistic understanding of metabolism in the whole human body. For all these challenges we have been using modeling and GEMs, but the outcomes will significantly improve when we combine them with the use of artificial intelligence (AI).

GEMs developed for protein secretion in yeast (Li *et al.*, 2022a), CHO cells (Gutierrez *et al.*, 2020), and human cells (Feizi *et al.*, 2017; Robinson *et al.*, 2019) are already being used to design cell factories that can produce produce a wide range of recombinant proteins more efficiently and at low cost. However, even though we

have considerable knowledge of the protein secretory pathway in these organisms, there are significant gaps in our understanding: the combinatorial space for engineering many target proteins involved in this pathway makes it difficult to find optimal design strategies but we predict that AI can improve the effectiveness in target identification. With the development of ecGEMs, we have already demonstrated optimal cell factory designs that can be used for the production of a range of different chemicals. However, ecGEMs rely on kinetic parameters, that is, turnover number, that in many cases are unknown. Additionally, many enzymes are catalytically promiscuous or reversible, resulting in reactions that lead to undesirable by-products or degradation of the product of interest. To address this challenge, we have built an AI model that helped us obtain $k_{cat}$ values for all enzymes present in more than 350 yeast species (Li *et al.*, 2022b). These parameters could be then used to populate functional ecGEMs for all these microbial species. We have democratized this approach by establishing an open-access database, GotEnzymes, that holds estimates of $k_{cat}$ values for more than 25 million enzyme-compound pairs across 8,099 organisms (Li *et al.*, 2023). We also created an AI model to map the so-called under-ground metabolism, that is, promiscuous functions of enzymes. The model trained on known enzyme-compound inter-actions, and we identified about 15,000 new reactions in yeast, that produce 15,873 new metabolites, of which the majority are engaged in lipid metabolism (Wu *et al.*, 2024). This demonstrated that metabolism is much more diverse than captured by traditional GEMs. Using the model, it was possible to identify many metabol-ites that can be formed as by-products, and this can now be used to guide the design of less promiscuous enzymes in metabolism that will improve biotechnological-based production. In the future AI could enable the development of better ecGEMs, but also enable the combination of model simulations with large experimental data obtained from large research programs using biofoundries, for even stronger prediction for optimized cell factory designs.

In the area of human health AI will also play an important role in the future. So far, most applications have been for image processing and more recently for protein structure predictions, provided by AlphaFold (Jumper *et al.*, 2021), for which Demis Hassabis and John Jumper received half of the 2024 Nobel Prize in Chemistry. Going further using AI for novel biologics design is, however, more challenging as we are dealing with an almost infinite number of combinations of amino acid sequences, with a desired targeted structure and properties. Breakthroughs are initially likely going to come through the design of smaller peptides or through the evaluation of small molecule-protein interactions, where AI is very well suited (Watson *et al.*, 2023), and for which David Baker received the other half of the 2024 Nobel Prize in Chemistry. For analysis of human metabolism and identification of novel drugs that target metabolism, the combination of GEMs with AI may be useful as GEMs can be used to generate large datasets that can then be used to train new AI models. For this there is a need for whole-body metabolic models that are built based on GEMs for different organs combined with a model describing blood circulation. With such models, the use of GEMs will be integrated into any drug development strategy as it will enable reduced experimental costs and reduced use of experimental animals, and such comprehensive models are very well suited to combine with AI for drug discovery.

In conclusion, we foresee that the application of GEMs, both alone and in combination with AI, will enable the provision of many new solutions that can improve both human and planetary health.

**Open peer review.** To view the open peer review materials for this article, please visit http://doi.org/10.1017/qrd.2025.1.

**Data availability statement.** Do not apply as no new data were generated in connection with this paper.

**Acknowledgments.** We would like to acknowledge all our fantastic students, postdocs, and research collaborators over the last 30 years.

**Author contribution.** J.N. and D.P. designed the structure of the manuscript. J.N. wrote the manuscript. D.P. edited and revised the manuscript.

**Financial support.** This work was financially supported by the Novo Nordisk Foundation (Grant No. NNF20CC0035580).

**Competing interest.** J.N. is a shareholder of Elypta AB, Melt & Marble AB, and Chrysea Inc. The remaining author declares none.

**Ethics statement.** Do not apply as no new research was conducted in con-nection with this paper.

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
