## [Reviewer Report]

Manuscript ID: QRBD-2024-0038

Overarching Comments

1. Request for greater clarity on what is being modeled

In many instances, the authors are referencing GEMs but do not always articulate what these GEMs are modeling. Is it flux-based analysis, optimization of media conditions/substrates for producing a specific product of interest, etc.? Additional clarity on this question would help the reader understand the real opportunities in applying these GEM approaches.

2. Do these models attain sufficient coverage of the metabolic reaction landscape?

By one estimate, yeast-GEMs have 4063 reactions, 2744 metabolites, and 1160 genes (https://metabolicatlas.org/explore/Yeast-GEM). It is not clear to this reader whether this represents a majority of metabolic reactions in yeast, and I suspect that this still significantly undercovers the yeast metabolic reactions. Can the authors comment on what proportion of the metabolic reaction space they estimate is covered by these modeling approaches?

Specific Comments

1. The authors state the following on page 2: “Today we consider metabolism to be almost completely mapped as we know most of the chemical entities and hence reactions that are occurring within living cells, but we don’t always know the identity and characteristics of the enzymes carrying out the reactions”

On what basis are the authors making this claim? Our lack of understanding extends beyond needing to know how the enzymes function. There are many “orphan metabolites”, which may be identified but carry out unknown function.

2. On page 2, the authors claim that the difficulty of engineering host metabolism is due to the efficiency “A major reason for this has been that it is often difficult to engineer the host metabolism in a way that uses most of the carbon atoms (from substrate) for production of the chemical/compound of interest.”

While the authors subsequently state that the underlying reasons for this poor efficiency is the (1) ‘hairball’ interactions between metabolites and enzymes and (2) extensive regulation of flux, there is a logical gap missing. Why do either of these reasons account for the inefficiency in metabolically engineering synthesis of a compound from carbon-based substrates?

3. On page 5, the authors state “Lactic acid bacteria cannot tolerate low pH and they require supplementation of complex feedstocks, and this makes it expensive and difficult to obtain pure lactic acid from these fermentation.”

As lactic acid bacteria can withstand low pH, thereby outcompeting other bacteria, the authors statement may not be correct.

4. At different points in the manuscript (page 8), it is pointed out that enzyme constrained GEMs have higher predictive strength.

As it is never clearly specified what are these models predicting, it is difficult to understand what makes these models superior to prior GEMs.

5. These findings are now taken forward in clinical trials to validate a biomarker approved for detection of recurrence in ccRCC patients from both urine and blood (pg. 9).

Please cite the clinical trial.

6. On page 10 “only few studies where causality between the human gut microbiome composition and disease development has been identified. The strongest data are in the field of cancer treatment with immune therapies where we have shown that the

presence of specific bacteria increases the response to treatment with check point inhibitors (Limeta et al., 2020).”

The authors may want to re-evaluate these claims. Even more than cancer immunotherapy, there is strong evidence to show that gut microbiome causally alters intestinal inflammation in colitis. An example reference is included below.

Zhu, W., M.G. Winter, M.X. Byndloss, L. Spiga, B.A. Duerkop, E.R. Hughes, L. Büttner, E. de Lima Romão, C.L. Behrendt, C.A. Lopez, et al 2018. Precision editing of the gut microbiota ameliorates colitis. Nature. 553:208–211. https://doi.org/10.1038/nature25172

Finally, let me express disbelief in the statement made early in the manuscript that almost all metabolic pathways have already been recognized.

---

## [Reviewer Report]

I enjoyed reading the review/perspective article by Nielsen and Petranovic. The manuscript provides a succinct overview of the vast field of metabolism and metabolic engineering. The historical and economic facts are provided throughout which keep the reader engaged as well as provides a broader context. There are a few minor errors/typos that need to be fixed:

1. Abstract: “…catalyzed by enzymes, co-factors and metabolites” -> this reads as if reactions are catalysed by metabolites. While this may be the case for few reactions, this is not general and usually not relevant at fast microbial kinetics.

2. Abstract: consider replacing “free energy” with ‘energetically favourable environment’

3. Introduction: consider avoiding “wizards”

4. Page 2, why metabolic engineering is difficult: The main reason is missing here – trade-off with growth, which is the evolutionary driving force that has shaped metabolism and regulation and thus it is difficult to drive fluxes away from growth to production.

5. Page 3: consider a paragraph break before “Following these initial models”

6. Page 7, last paragraph: consider changing “obtaining” to predicting or describing since the sentence refers to models.

7. Page 10: the latest estimates for number of microbial cells (not counting viruses though) in the human body suggests that it is in the same order as human cells.

---

## [Editor Report]

Congratulations to very nice paper! On my screen I has problems some symbols in figures jumping around. I am sure you can revise the illustration material to be as you wish it, when communicating with the production department.